# Promoting Equity in Access to Quality Early Childhood Education in China

**DOI:** 10.3390/children10101674

**Published:** 2023-10-10

**Authors:** Nirmala Rao, Yi Yang, Yufen Su, Caroline Cohrssen

**Affiliations:** 1Faculty of Education, The University of Hong Kong, Hong Kong, China; u3004256@connect.hku.hk; 2Department of Education Studies, Hong Kong Baptist University, Hong Kong, China; yiyang@hkbu.edu.hk; 3School of Education, University of New England, Armidale, NSW 2350, Australia; ccohrsse@une.edu.au

**Keywords:** early childhood education, accessibility, quality, equity, China

## Abstract

This paper reviews the Chinese government’s efforts and progress towards ensuring equitable access to quality early childhood education (ECE). It begins with an overview of the Chinese context and analysing the reasons for a policy focus on ECE in recent decades. Thereafter, from a historical perspective, it critically reviews policies pertaining to enhancing access to and the quality of ECE in China since 2010. Nationally representative data are used to document policy implementation. Our analyses of national policies and representative data suggest that the state considers ECE a public good. This is evident from policy changes, efforts to bridge urban–rural disparities, fiscal allocations to the most economically disadvantaged groups, and strategies to enhance the quality of ECE.A significant focus on policy implementation with improved regulation and monitoring of services provided by both state and non-state actors was found. However, it is evident that gaps persist between urban and rural areas regarding infrastructure and resourcing, gross enrolment rates, and teacher–child ratios. That stated, the momentum that has driven policy change and the dramatic gains demonstrates the priority accorded to ECE.

## 1. The Chinese Context

China is the world’s second most populous country, with 1.4 billion people living in 31 provinces [1]. China has the second largest number of children worldwide, but the population is declining. In 2023, there were 61,144,621 children under the age of five [2]. Addressing the differing needs of children with diverse backgrounds by providing high-quality ECE services nationwide is a formidable task.

In addition to the current cohort size, other contextual factors should, in theory, affect the future demand for and supply of ECE. One would expect that moving from a one-child policy to a universal two-child policy in 2016 and to a three-child policy in 2021 would have led to an increase in the birth rate. This did not happen as the birth rate continues to decrease in China. Social factors, however, have led to an increased demand for high-quality ECE services. There has been increased competition in China to secure admission to elite universities, and parents want high-quality ECE as they do not want children to be ‘behind at the starting gate’ of school. Many Chinese parents do not want more than one child because of the increased competition for admission to elite universities and the increasingly unaffordable cost of raising a child. In response, in 2021, the Chinese government issued *Opinions on Further Reducing the Work Burden of Students in Compulsory Education and the Burden of Off-campus Training*, colloquially known as the Double Reduction policy. The Double Reduction policy aims to reduce excessive parental pressure and has been deemed one of China’s responses to the dropping birth rate and the potential population crisis [3]. It may take some time for the impact on ECE of the Double Reduction policy to be seen. Further policy interventions may also increase the birth rate and the demand for ECE in the next decade. However, it is clear that China’s educational policies, discussed below, acknowledge the importance of investing in the early years to prepare children for a high-tech knowledge-intensive economy.

Economic development in China is vastly uneven. Rural areas and provinces located in the western and central regions of the country have benefited less from economic growth than more-developed urban areas and the eastern regions of the country. Rapid urbanisation has taken place: in 2021, China’s urbanisation rate was 64.7%; in 2000, the urbanisation rate was 36%. This rapid urbanisation, together with uneven regional development, has spurred great internal migration in China. More than 240 million people have migrated, primarily from rural areas to urban areas and from less-developed provinces to more-developed provinces [4]. While migrating to urban areas may mitigate economic hardship, migrant populations are vulnerable to job insecurity and poor living conditions [5]. More importantly, migration has also led to challenging childcare issues.

The latest statistics suggested that 1 in 3 children in urban areas are migrant children, and 4 out of 10 children in rural areas are left behind in rural areas when their parents move to urban areas for employment [6]. On the one hand, migrating to urban cities with parents may benefit children since it mitigates issues associated with separation from parents, possibly reduces the negative impacts of growing up in poverty, and allows children to accrue advantages related to living in more-developed cities [7]. On the other hand, it is difficult for migrant children to assimilate successfully into local communities due to institutional practices and policies such as *hukou*. A *hukou* book is a legal document associated with China’s household registration system. The *hukou* is based on the birthplace of the head of the household. Citizens can have an urban or a rural *hukou*; these have different benefits, including access to government jobs, healthcare, and public education. Rural to urban migrant workers hold a rural *hukou* that limits their own and their children’s access to resources, including public education. Highly skilled migrant workers may be granted an urban *hukou* to attract talent to urban areas. In 2016, the State Council issued a policy indicating the intention to grant urban *hukou* to 100 million rural migrants by 2020. As children’s admission to the public school system is tied to their *hukou* status, it is difficult for migrant children with a rural *hukou* to secure a place in a mainstream public kindergarten in urban areas, even when they live with their parents in these urban areas. ECE services for migrant children have been of great concern, especially in metropolitan cities that attract the most migrant workers, such as Shanghai, Guangzhou, Shenzhen, and Beijing. The national government has attached great importance to ECE provision for this group and has urged local governments to create more opportunities to ensure ECE access for migrant children. Nonetheless, implementation of the policy varies widely across the country as it depends on local governments’ fiscal capacity and commitment.

In China, ECE refers to organised and centre-based education and care services for children aged from birth to 6 years. Nurseries (childcare centres) provide services for children below three years of age; full-day kindergartens generally serve children from 3 to 6 years. Children typically spend 7 to 8 h per day in kindergartens. Children typically attend three levels of age-segregated classes (kindergarten 1, kindergarten 2, and kindergarten 3), although some kindergartens have mixed-age classes. A one-year pre-primary education programme may also be provided within some primary schools for children from 5 to 6/7 years to utilise the limited resources in remote rural areas fully. Publicly funded kindergartens (hereafter, referred to as public kindergartens) are typically the preferred option for most parents, but in urban areas, competition for a place in renowned public kindergartens is very keen. This situation differs from Hong Kong where all ECE services are provided by the private sector. In Hong Kong, the government provides subsidies and monitors the quality of ECE provision. Children in China with special education needs are either enrolled in mainstream classrooms or in schools that specifically cater for children with additional needs. This paper uses the term ECE to refer to centre-based kindergarten education unless otherwise specified.

In 2010, the *Outline of China’s National Plan for Medium and Long-term Education Reform and Development 2010–2020* [8] (hereafter, the Outline) was a watershed moment in the history of ECE. Since 2010, government policy initiatives and increased financial investment have focused on enhancing the supply and quality of ECE. Extant studies have examined the momentum toward accessible and high-quality ECE [9,10,11,12]. Many have considered the effectiveness of specific policy initiatives and programs [13,14], as well as national guidelines on teaching and learning [15,16]. Yet, the review of how China has enhanced accessibility, quality, and equity has been less thorough. This paper reviews the Chinese government’s efforts and progress towards ensuring equitable access to quality ECE. Policy documents relevant to ECE were analysed to facilitate a comprehensive understanding of China’s strategies to promote ECE since 2010. Publicly available datasets were examined to determine the impact of policy initiatives in this context.

## 2. The Changing Role of the State in the Provision and Regulation of Early Childhood Education

Debate on the degree to which the state should be involved in ECE is ongoing. One perspective is that responsibility for ECE should be shared by parents and private sector ECE providers. Another perspective is that ECE is a public good that benefits society and should be accessible to all. The latter view requires the state to be the major supplier of ECE and to assume the responsibility of monitoring service provision. The debate about the role of the state in ECE guided our discussion and analysis of the Chinese government’s involvement in ECE.

ECE in China has undergone a significant shift with progressively increasing involvement of the state in the provision and regulation of ECE. In the 1990s, with the development of the market economy, the government introduced the “government retreats but private-sector advances” policy, with which the government gradually reduced public funding for ECE and encouraged non-state sectors (non-state actors: individuals and organisations, ranging from home-schooling providers to non-governmental institutions, that provide education and care services for children; private providers are a major subset of non-state actors) to provide ECE services. Since then, many public kindergartens have been closed, suspended, merged, or become privately owned. The non-state sector expanded quickly and gradually became the predominant provider of ECE services (68% were run by non-state actors by 2010), and the non-state providers shoulder the major responsibility of ECE provision. By this time, the demand for ECE vastly exceeded supply, resulting in a 56.6% GER in ECE for 3- to 6-year-olds and access to ECE varied considerably across provinces and urban–rural areas [17]. Furthermore, most private kindergartens were for-profit, more expensive, and had lower levels of quality than public kindergartens. Chinese parents faced significant challenges in enrolling their children in high-quality, affordable kindergartens during this period.

The 2010 Outline was the first time that ECE had been included in the government’s strategic plan and signalled the start of national investment in ECE as a public good after decades of market-driven ECE [10]. In the Outline, the Chinese government put forth objectives and ambitious goals to improve the ECE system. Three key objectives were specified: (1) universal access to ECE; (2) increased government responsibility in the provision of ECE; and (3) improved ECE in rural areas. Since then, there has been sustained government focus on ECE.

### 2.1. Development of ECE Services through National Policies

Following the Outline, the State Council launched *Suggestions Concerning Current Development of Early Childhood Education* [18] (hereafter referred to as the Suggestions), which outlines the implementation strategy to meet their objectives. These Suggestions make a case for ECE and contain ten specific strategies to achieve the goals specified in the Outline. Implementation strategies included expanding the supply of ECE, supporting private kindergartens, and supporting the construction of kindergartens in rural areas. Other strategies concerned the ECE workforce, increasing public funding for ECE, health and safety issues, improving childcare and educational practices, multi-sectorial collaboration, and government leadership.

A notable recommendation in the Suggestions was to evaluate progress via a three-year action plan. Since the introduction of the Suggestions, county-level local governments or above have been required to identify challenges in the provision and quality of ECE services within their administrative units, set goals to address these challenges, take actions, and monitor progress. Formal reports about their progress are made to a higher administrative unit of the government every three years. The action plan ensures that responsibilities are assigned precisely and that implementation of the policies is tracked regularly, thus increasing the likelihood of achieving the strategic objectives. The Outline and Suggestions have played pivotal roles in developing other strategic plans and policies focusing on specific ECE areas or groups of children. For example, the concrete goals regarding ECE enrolment rates were re-emphasised in the Child Development Outline of China (2011–2020), in national poverty alleviation plans, and in educational plans for certain groups of children, such as children with disabilities.

In the past decade, policymakers, researchers, and other stakeholders in China have been lobbying for an ECE law to ensure children’s right to education, especially for those most likely excluded. The ECE law (draft) includes stipulations on children’s rights; the responsibilities of the state, families, and communities, and regulations related to the operation of kindergartens. After soliciting opinions and consultations, the ECE law (draft) entered the legislative process in 2018. The ECE law (draft) was approved at the Executive Meeting of the State Council in June 2023 and was passed on to the National People’s Congress for final approval. The legally binding ECE law (draft), once passed, will further direct resource allocation to promote social justice and quality and underscore the responsibilities of the central, provincial, country, and local educational authorities. The legal framework demonstrates the Chinese government’s commitment to ensuring that children are able to access ECE.

### 2.2. Increase in Public Expenditure on ECE to Reach Disadvantaged Children

Two major drivers have contributed to an increase in public expenditure on ECE to reach disadvantaged children in China. First, the Convention on the Rights of the Child (CRC; United Nations, 1989) specifies that all children regardless of their gender, nationality, ethnicity, region of residence, and family background have the right to education and care. The state is obligated to ensure children’s right to ECE. China signed the CRC in 1990 and ratified it in 1992. Second, economic and educational sciences have shown that investment in ECE can yield substantial returns for individuals’ life-long development and countries’ human capital development [19,20].

With the recognition of the importance of ECE for human and societal development, the Chinese government has increased investment in ECE. Public funding is the major source of investment in ECE with limited investment from families [21]. From 2011 to 2020, the Chinese government invested over CNY 170 billion (USD 23 billion) with an average of 20.6% yearly growth in investment in ECE. The proportion of the total education expenditure allocated to ECE also increased from 2.2% in 2011 to 5.9% in 2020 [21]. Yet, by 2017, state funding for ECE was less than 0.2% of the nation’s GDP, which was lower than that for primary education (1.22–1.39%) and the OECD average (0.6–0.8%) [10].

To move toward equal access to ECE, the government has allocated more public funding to less economically developed provinces and rural areas. As noted earlier, less economic development in western and central regions reduced local governments’ ability to invest in ECE; this, in turn, resulted in lower levels of kindergarten availability and enrolment compared with those in more-developed provinces in the eastern region [13]. Within each province, local governments also prioritised the allocation of public funding to urban kindergartens [22]. Thus, it is not surprising to see substantial regional and urban–rural disparities in ECE enrolment. Since 2010, the national government has adopted a targeted approach, allocating financial resources to the less economically developed provinces in the western and central regions. Most of these funds were channelled into building kindergartens and providing subsidies for children in less-developed provinces who may otherwise be unable to access ECE. With this targeted support, the pace of ECE development was fastest in the western and rural areas. From 2011 to 2020, 80% of newly established kindergartens were located in the central and western regions and roughly 60% were located in rural areas [21].

Within provinces, local governments are required to allocate a significant proportion of public funding to ECE in rural areas. As shown in Figure 1, local governments distributed higher proportions of public financing to ECE in rural areas in almost all provinces in 2018 compared to 2011. In 2018, public funding for ECE, including funding provided by the central government and by local governments, was CNY 177 billion, or approximately 48.3% of the total ECE fiscal revenue; around half (49.27%) of this public funding was distributed to rural ECE, compared with 33.88% in 2010. Among the 31 provinces (municipalities or autonomous regions), 19 allocated more than half of the public ECE funding to rural kindergartens, with the rural allocations being highest in the Qinghai (78.82%), Gansu (77.15%), Xinjiang (75.47%), Jiangxi (71.76%), and Hebei (70.33%) provinces.

### 2.3. Coordination of Non-State Actors and Families

In addition to the state’s expanding role in policymaking and funding allocation, the Chinese government regulates non-state actors to build a sustainable, vibrant, and flexible ECE system, encouraging non-state stakeholders to provide ECE services that suit families’ diverse needs. For example, the Chinese government required property developers to build at least one affordable kindergarten in newly built residential communities. With increased demands for high-quality childcare but limited available supply, the Chinese government mobilised non-state actors to provide flexible childcare services, including community-based childcare, home-based childcare, and full-day, half-day, or drop-in day care programmes to meet families’ needs [23].

Each province also formulates policies to support non-state actors to provide childcare services. For example, Suzhou, a city in Jiangsu Province, provides childcare centres with a construction subsidy of CNY 10,000 per childcare place and an operating subsidy of CNY 300–800 per child per month for inclusive childcare; Wenzhou, a city in Zhejiang Province, provides childcare centres with subsidies at the rate of 1.2 to 2 times that of local kindergarten per pupil fees. To further alleviate parents’ financial burden, the Chinese government introduced a tax exemption policy for parents with children under the age of three who can enjoy CNY 1000 monthly tax exemption for each family.

## 3. Expanding ECE Accessibility: From Availability to Affordability and Suitability

Access to ECE can refer to the supply/availability of ECE or to children’s participation in ECE programmes. Indeed, ECE access is a multi-dimensional concept encompassing factors relating to availability, usage, cost, transportation, language, and cultural competence [24]. A comprehensive conception of access to ECE should consider (i) demand (what parents want); (ii) supply (what is available and affordable); (iii) how well ECE programmes meet the needs of children and their families; and (iv) how many children attend high-quality ECE programmes [24]. Data on most of these access indicators are not publicly available in China, making it difficult to analyse children’s participation in high-quality ECE settings. Using open-access administrative data, we found a policy focus moving from ECE availability and participation toward ECE affordability and suitability, with an increasing awareness of the education and care for children under three years of age.

### 3.1. ECE Availability and Participation

In 2010, the Outline and the Suggestions ushered in a new wave of ECE reform, where new policies were implemented to address the challenges of ECE access and cost. The long-term goal was to universalise access to ECE services for all 3- to 6-year-olds by 2020, ensuring that all 3- to 6-year-old children could attend kindergartens with affordable tuition fees. To achieve this goal, local governments have been striving to increase the number of private kindergartens that charge tuition fees comparable with those charged by public kindergartens. Another primary focus has been the promotion of equity, typically pursued by prioritising rural areas and less economically developed provinces where ECE enrolment is lower than the national average [22].

Available data demonstrate the success of policies and strategic plans to increase ECE availability. Figure 2 showcases the remarkable increases in the number of kindergartens and the GER in ECE for 3- to 6-year-olds. Between 2011 and 2021, more than 128,000 new kindergartens—a rise of 76.8%—were established. Latest official reports suggest that 289,200 kindergartens served over 46 million 3- to 6-year-olds in 2022. Meanwhile, the GER of the 3- to 6-year-old children’s ECE was 89.7% in 2022 [25].

### 3.2. ECE Affordability and Suitability

Through boosting the number of kindergartens, the state has sought to further ameliorate the issue of the high costs of ECE programmes with an innovative strategy: promoting Pu Hui Xing kindergartens. Pu Hui Xing kindergartens, translated as ‘affordable kindergartens’, charge affordable tuition fees for all families, including all public kindergartens and Pu Hui Xing private kindergartens [26]. Tuition fees of public kindergartens are determined by the local governments based on their merits in a local quality rating scheme. Public kindergartens with higher ratings charge slightly higher tuition fees than those with lower ratings. Public kindergartens generally charge lower tuition fees than private kindergartens, including Pu Hui Xing private kindergartens. Pu Hui Xing private kindergartens are required to follow the tuition fees prescribed by local governments. In return, local governments provide Pu Hui Xing private kindergartens with subsidies such as lower rental costs. Private kindergartens that remain for-profit retain the autonomy to decide fees but have profit caps set by the government in some cases. At the same time, local governments limit the number of for-profit private kindergartens to ensure Pu Hui Xing kindergartens are the main ECE providers. For example, the Jiangsu provincial government stipulated that the profit–cost ratio should not exceed 15% [27]. These measures were all aimed at increasing the affordability of ECE.

Administrative statistics suggest that by the end of 2022, 85% of all kindergartens were Pu Hui Xing kindergartens, already achieving the national government’s target of 85% of affordable kindergartens by 2025 [25]. Among all the Pu Hui Xing kindergartens, 52% were public-funded, showcasing the state’s tremendous effort, with an increase of 149.7% in the number of public kindergartens compared with 2011 [21]. A study of 2928 private kindergartens in 17 provinces revealed that over 75% of the surveyed for-profit private kindergartens intended to or had transformed into Pu Hui Xing kindergartens [27].

The expansion of Pu Hui Xing kindergartens has enhanced ECE availability and affordability. However, tuition fees continue to be a significant financial burden for low-income families. Although local governments make a substantial financial contribution to Pu Hui Xing kindergartens, a reasonable and defined cost-sharing formula that allocates costs among the government, kindergarten operators, and parents has yet to be developed.

## 4. Regulating ECE Quality: From Structural Quality to Process Quality

ECE is not included in the compulsory education system. With the rapid expansion of kindergartens, the government realised that access alone could not necessarily promote the learning and development outcomes of children. Hence, concerted efforts have been made to improve ECE quality. There are two main aspects of preschool quality: structural and process. On the one hand, structural quality refers to the physical setting, class size, teacher–child ratio, teacher qualifications, the availability of appropriate materials and equipment, and the safety and cleanliness of the setting. Process quality refers to the quality of the teacher–child interactions and the extent to which teachers provide stimulating and developmentally appropriate activities.

In the initial stage of expanding ECE services, the Chinese government focused on improving kindergartens’ structural quality, such as infrastructure, equipment, facilities, and materials. This was for two reasons. First, structural elements are easy to define and regulate. Second, the environment plays a vital role in children’s learning and development within traditional Chinese philosophy. One of the best examples to illustrate how Chinese philosophy emphasizes the importance of environment comes from the legend of Mencius. The mother of Mencius, a widow, moved her residence three times until she found the right location to raise her child. This was next door to a school. A favourable environment is assumed to promote proper and moral behaviours, while a chaotic and disorganised environment does not lead to propriety and effective learning. Thus, the kindergarten quality rating guidelines emphasize a structured learning environment with stimulating learning materials. Notably, with the gradual improvement in kindergarten structural quality, the Chinese government has introduced specific guidelines and regulations to promote process quality.

Supporting play-based curricula and teacher–child interactions have been consistently highlighted in national policies. Along with specific quality indicators and guidelines, policy attention is increasingly focusing on building system capacity to monitor and evaluate quality provision. In this section, we present the Chinese government’s endeavours to improve ECE quality from the indicators of ECE quality evaluation to the quality assurance mechanism.

### 4.1. Shifting Attention from Structural Quality to Process Quality

The structural quality indicators for kindergartens include tuition fees, physical settings, space and facilities, class size, teacher–child ratios, and staff qualifications, as well as health, hygiene, and safety. The process quality indicators mainly focus on teaching, learning, and child development.

**Tuition fees.** The Chinese government has set strict tuition fee limits for public and private kindergartens that receive public funding. Local governments monitor the tuition fees charged by these kindergartens. The tuition fees for public kindergartens are set by provincial education departments based on the local economic development level [28]. Non-profit private kindergartens must follow the same standards as public kindergartens in the same district. For-profit private kindergartens are given the autonomy to determine their tuition policies, but governments restrict their profits to prevent excessive profit-driven behaviours. Although these policies help prevent kindergartens from overcharging, the efforts have been criticised for being driven more by regulatory goals than by a desire to promote high-quality services by establishing a rational cost-share system [14]. Indeed, the standards are similar across kindergartens and have yet to consider the variations in operational costs and the public funding received by different kindergartens [14]. For example, within the same district, non-profit kindergartens typically receive less public funding support than their public counterparts, and the strict tuition fee regulation could result in reduced quality in order to improve profits.

**Physical settings, space and facilities.** The Kindergarten Construction Standards contain recommendations regarding where kindergartens should be located, how they should be built, and other aspects relating to the physical setting. It is essential to adhere to these standards when constructing new kindergartens. For instance, kindergartens should have at least 4 square meters of outdoor playground per student [29], and at least half of the outdoor play area should be exposed to sunlight for at least 2 h on a winter’s day. This is aligned with the requirement outlined in the Kindergarten Working Regulations that kindergartens should ensure children’s participation in outdoor activities for no less than 2 h or more per day [30]. This is because Chinese kindergartens accord importance to outdoor physical activities and consider that exercise and exposing children to sunlight benefits their health and well-being. Standards related to indoor space per student are also specified in the Kindergarten Working Regulations.

**Class size, teacher–child ratios, and staff qualifications.** Class size and teacher–child ratios are indicators of ECE structural quality, which are considered a prerequisite of quality and stimulating teacher–child interaction [31]. A reasonable class size and teacher–child ratio can provide more opportunities for individualized teacher–child interaction and support. The Ministry of Education (MoE) has mandated that the number of children in a classroom should be at most 25, 30, and 35 for classrooms serving children aged 3–4 years, 4–5 years, and 5–6 years, respectively [30]. Furthermore, each classroom should have at least two teaching staff and one caregiver [31]. However, in practice, class size and teacher–child ratios vary widely across kindergartens, with many kindergartens experiencing difficulty in complying with government regulations. Staff qualifications are strongly emphasised in different documents, and licensing requires proof of staff qualifications. Kindergarten principals are required to hold at least an associate degree, have three years of working experience in the field, be able to take leadership roles, and complete a kindergarten principal pre-job training course. Teachers must complete at least a post-secondary ECE teacher preparation programme and obtain the ECE teacher certificate. At the same time, caregivers should hold at least a high school diploma. Qualifications of healthcare providers (i.e., doctors, nurses, and healthcare-related workers) are also specified in relevant policies. Although these minimum requirements have been criticised as too low to ensure high-quality service, some kindergarten staff in poor, rural areas are still working towards meeting these minimum requirements.

**Health, hygiene and safety.** Ensuring children’s healthy development is the joint responsibility of kindergartens, local Departments of Health, and other institutions, such as the Centre for Disease Control and Prevention. The National Health Commission and the MoE jointly regulate kindergarten health, hygiene, nutrition, and safety issues. The Ministry of Health develops relevant policies and guides kindergartens on how to implement and monitor prescribed practices. The local Departments of Health are responsible for evaluating the health, hygiene, and safety conditions of newly established kindergartens. A full-time certified healthcare provider is also a requirement, especially for kindergartens serving 150 or more children. Kindergartens are required to develop a comprehensive school-based healthcare policy and to conduct daily physical check-ups for children, monitor child growth and development, implement infectious disease prevention checks, set up daily sound routines, conduct hygiene practices, and provide balanced diets for children. These practices help prevent and reduce the prevalence of common diseases in young children and promote child development and well-being [32].

**Teaching, learning, and child development.** Unlike structural features for which the government has, as we have seen, detailed requirements for teaching and learning, the critical process quality indicators are less clearly defined. Local governments have autonomy to develop their own tools for assessing teaching and learning quality. However, the MoE provides general guidance on planning and implementing educational programs. For example, the Kindergarten Working Regulations highlight that kindergarten educational practices should be developmentally appropriate and respect developmental differences in individual children. They also identified play as the primary pedagogical approach and require kindergartens to create a rich and stimulating learning environment to support children’s learning. Teaching literacy (including pinyin) is strictly prohibited in kindergarten classrooms. In 2022, the MoE issued the first national guidelines that explicitly stipulate critical dimensions of process quality: Evaluation Guidelines for Early Childhood Education and Care in Kindergartens. The guidelines address teaching and learning processes, learning environment settings, and teaching workforce. It is worth noting that in the teaching and learning processes dimension, the policy affirms the importance of activity organisation (Chinese policy avoids the use of the word ‘class’ as it implies academic orientation in the Chinese context), teacher–child interactions, and home–school–community collaboration. A child-centred and play-based curriculum that integrates many hands-on exploration opportunities that tap into children’s daily experience is considered high-quality ECE provision. The most common curriculum model in Chinese kindergarten is theme-based and blends five domains of learning and development. These include health and nutrition, language and literacy, numeracy and science, socioemotional well-being, and artistic skills. Kindergartens and schools are encouraged to adapt existing curriculum models (e.g., the Reggio Emilia approach and High-Scope curriculum) to develop their school-based curriculum to align with government guidelines. One such example is the Anji Play approach, which encourages risky play in natural environments. Some provincial governments have developed a local curriculum that works as the reference for school-based curriculum implementation.

The shift in attention from structural to process aspects of ECE quality reflects research that suggests that structural quality alone does not guarantee stimulating learning experiences or positive learning outcomes for children [33,34]. Thus, promoting ECE teachers’ professional learning to enhance process quality has also been a priority. Indeed, supporting ECE teachers’ professional learning is positioned as a critical pathway for quality improvement in the Chinese government’s Evaluation Guidelines for Early Childhood Education and Care in Kindergartens. However, despite the fact that ECE teacher well-being is closely associated with children’s socioemotional development [35], ECE teachers’ socioemotional well-being is receiving less policy attention.

Although child development outcomes are viewed as a critical indicator of service quality [36], they are not used for national ECE quality evaluation in China. Child development is not assessed using standardised population-level monitoring tools, such as the Early Development Index in Canada or the Australian Early Development Census (formerly the Australian Early Development Index). This is intended to avoid bringing academic or performance pressure to bear on the play-based curriculum, kindergartens, and teachers. Many kindergarten teachers in China refer to age-specific desirable outcomes for child development listed in the guidelines as a formative assessment tool to assess children’s competencies. Assessment results may be used to adjust the level of educational activities for children [11]. In some provinces, the government may also include desirable outcomes for child development in their quality evaluation tools as an outcome indicator of ECE quality.

### 4.2. Emerging Quality Assurance and Enhancement Mechanisms

Globally, there is increasing consensus that a strong monitoring and quality assurance system is pivotal to the sustainable development of ECE systems [15]. A monitoring and evaluation system can support governments to assess the efficacy of their ECE system and identify groups with particular needs in order to provide timely and targeted support. Although existing cross-national data, such as the Multiple Indicator Cluster Survey (MICS) and the Demographic and Health Survey (DHS), can provide international comparable information for policymakers in China, information that comprises country-relevant indicators are still needed.

China’s goal of universalising high-quality ECE education has provided the impetus for developing and implementing a national system to evaluate ECE quality [37]. Before 2010, kindergarten quality evaluation was overseen by local educational governments that used locally developed quality rating systems. The rapid expansion of ECE facilities in the past decade has led to increased urgency in establishing a quality assurance framework to monitor ECE quality across the country. It should be noted that there have been some marked changes in the kindergarten quality evaluation system over the past ten years. In the past, the monitoring of kindergarten quality by local authorities was not mandatory, and a large number of kindergartens were not evaluated [16,37]. The monitoring of ECE quality by local authorities is no longer optional and is now required. All kindergartens must be evaluated, not just a few “model” kindergartens (see Demonstration Kindergarten Scheme below). Since 2022, local governments must comply with the national guidelines—Evaluation Guidelines Early Childhood Education and Care in Kindergartens—in terms of evaluation modality, frequency, and indicators to come up with their own evaluation plans.

#### 4.2.1. The Demonstration Kindergarten Scheme and the Kindergarten Quality Rating System

The evaluation of kindergarten quality since the 1950s has primarily followed the demonstration kindergarten approach that identifies high-quality kindergartens capable of being models of good practices for other kindergartens and supporting the latter to improve their quality [38]. To assess whether or not a kindergarten is qualified to serve as a model, local governments use the province-specific Demonstration Kindergarten Quality Rating System or Kindergarten Quality Rating Systems (QRSs) [37]. These two monitoring systems have similar aims. Despite their shared objective of identifying high-quality kindergartens, quality rating contents and implementation vary substantially across provinces, municipalities, and autonomous regions. A QRS is the primary tool used to assess kindergartens regularly [39]. The Kindergarten QRS of Guangdong province, for example, relies on a 39-item quality evaluation tool to measure (i) some structural dimensions of quality (facilities, learning materials, class size, staff-to-child ratio, and staff qualification) and (ii) management of learning and teaching, family–kindergarten relations and finance [40]. Kindergartens that wish to be accredited as officially rated kindergartens must identify and address their areas of weakness to meet the quality standards. Incentives for earning accreditation are increased funding and the receipt of awards of excellence [40].

The validity and comprehensiveness of existing QRSs require further attention. First, as noted earlier, rating standards may vary substantially across provinces. Furthermore, a kindergarten rated as “excellent” in an economically less-developed western province in China may be of considerably lower quality than a kindergarten rated as excellent in the municipality of Shanghai. This reflects an absence of national benchmarks for ECE quality [39]. QRSs also tend to give more weight to structural elements of ECE quality than to indicators of process and outcome quality [16]. Moreover, even if a province-specific QRS effectively differentiates kindergartens of differing quality, the kindergartens that receive high rankings on the QRS do not score very high on standardised measures of observed quality [37]. Another issue, particularly in Pu Hui Xing private kindergartens, is that some local governments and kindergartens believe that it is impossible to provide services that are simultaneously universally affordable and of high quality. Local governments have not provided enough funding to transform kindergartens of low and mid-range quality into high-quality kindergartens [27].

There has been some concern that applying QRSs may have paradoxically increased disparities in ECE quality and inequality [16,37,41]. Some local governments have deliberately allocated more resources to support the successful accreditation of a limited number of kindergartens, mainly publicly funded kindergartens [38]. Accredited kindergartens are provided with more help, more support for teacher training, and more public funding, leading to inequalities in kindergarten quality [38]. As access to these high-quality kindergartens is limited, only a very small proportion of children succeed in securing admission to these kindergartens, often those with higher socioeconomic status backgrounds [37,38]. Hence, the Matthew effect may be a sequel to the QRS.

#### 4.2.2. Kindergarten Supervision and Evaluation Scheme

In 2017, the national government initiated the Kindergarten Supervision and Evaluation Scheme to evaluate kindergarten quality across the country to improve the former kindergarten evaluation system, to promote the local government’s supervision of kindergartens, and to improve ECE quality and effectiveness. With this scheme, nationwide kindergarten evaluation is gradually being standardised, allowing the government to collect data on pre-determined quality indicators. Provincial governments have been given autonomy to develop the measure and implement the evaluation, which generally includes five aspects (i.e., the condition of the kindergarten, safety, hygiene and healthcare, education and care, staff, and management).

The MoE requires each county (district) government to supervise and evaluate every kindergarten in their county at least once in five years. After the evaluation, local governments provide kindergartens with written feedback and asks them to improve in their areas of weakness. The results of the evaluation, indicated by a rating on a five-point Likert scale in some provinces, are publicly available and used for amending and implementing local ECE policies. In 2019, the MoE developed an online system for local governments to conduct evaluations, collect data, and thus build a national ECE quality database [42].

The tension between teacher autonomy and the quality monitoring system is always a challenge when local governments take the responsibility of monitoring and evaluation. Teachers may be overly concerned about meeting quality standards to pass government inspections, leaving them little room to practice their own educational philosophy [43]. Thus, it is vital to provide teachers with sufficient professional support through school-based or community-based professional collaboration. It was found that when kindergartens encouraged teachers to participate in school decision making and curriculum innovations, ECE teachers enjoyed more curriculum autonomy, which in turn, enhanced their job satisfaction and well-being even under a stringent quality assurance system [44].

## 5. ECE Equity: Diminishing Persisting Disparities in Access to High-Quality ECE

In this section, we review the extent to which disparities and barriers have been diminishing as a consequence of the policy effort to promote equitable ECE for all children.

### 5.1. ECE Enrolment

**Gender**. Data on sex ratios at birth indicate a strong son preference in China. In 2005, there were 119 boys born for every 100 girls, but this decreased to 112 boys for 100 girls in 2017 [45]. Official data from MoE indicate no differences between girls’ and boys’ ECE enrolment ratios [26]. This means that although the son preference is still prevalent, parents are equally likely to send boys and girls to ECE once the child is born.

**Age**. Children’s ECE participation in China varies with their age. As ECE services in China mainly target 3- to 6-year-olds, and 6 years is also the official enrolment age of primary grade 1, it is not surprising to see that children aged under three and those aged six and older accounted for a small proportion of the enrolled children (2% and 4%, respectively). Among the 3- to 5-year-olds, there were substantial age differences in ECE participation. The 3-year-olds accounted for only 26% of the enrolled children, although this percentage has increased since 2011. This figure also aligns with the current policies of prioritising older children in kindergarten enrolment. Notably, nationwide survey data showed a meagre enrolment ratio for 3-year-olds in rural areas, suggesting less time is spent in kindergartens by rural children [46].

**Programme type**. As mentioned earlier, kindergartens in China are run by for-profit entities or public sector entities, such as government departments, universities, the army, and neighbourhood communities. Public kindergartens are more likely to attract better educated and more competent teachers because the former offer better terms of service. Public kindergarten teachers are more likely to be awarded *Bianzhi* and *Zhicheng* status by local educational authorities. Bianzhi is similar to a tenured position, and *Zhicheng* is a professional rating based on the teacher’s education degree, professional performance, and work experience [47]. Bianzhi is primarily provided to teachers working in public kindergartens, while teachers working in private kindergartens may now apply for *Zhicheng*. These two factors significantly affect teacher remuneration since teachers with *Bianzhi* and *Zhicheng* receive better salaries and perquisites than those who do not have them [39]. As a result, it is not surprising to see public–private differences in teacher education and teacher-to-child ratio in favour of public kindergartens [47].

**Region/urbanicity/SES**. Positive changes in the GER of 3- to 6-year-olds in ECE across the nation have been documented in administrative data and household surveys. The government reported a more significant ECE expansion in the western region relative to the eastern and central regions [48]. Between 2010 and 2018, the number of enrolled pre-schoolers in the western region increased by 76.3% compared with 65.1% and 35.7% in the eastern and central areas, respectively [48]. The expansion in the western region was the result of the government’s intensified policy efforts and the low baseline GER in the region. Nationally, GER in urban and rural areas also increased by 26.6% and 54.6%, respectively [48].

Using nationwide household survey data, Su and colleagues compared kindergarten enrolment ratios for children aged 4–6 years in the rural western region with the urban eastern region and found that the enrolment gap between these two regions had narrowed from 36.36% in 2010 to 19.66% in 2016. Similarly, the gap in the enrolment ratio for children of mothers with tertiary education and those of mothers with no more than primary education reduced from 30.76% to 8.88% in the same period. Despite these positive changes, children, especially 3-year-olds, living in rural areas and those from families with lower SES are still less likely to attend kindergartens than other children [46]. Government data indicate a low GER in ECE in poverty-stricken areas and areas inhabited by predominantly ethnic minorities where public kindergartens were still unavailable [48]. These areas, some of which have a GER below 50%, are in need of intensified efforts to ensure access to ECE [48].

Public kindergartens charge lower tuition fees than private kindergartens, and the expansion in the number of public kindergartens has been most prominent in rural villages, where families may have been less likely to send their children to kindergartens due to economic constraints. Hence, there has been an increase in ECE enrolment in rural areas. Indeed, government data indicate that between 2010 and 2018, ECE participation rates increased by 61.6% in rural areas and 56.4% in urban areas. This is not surprising, as participation rates were notably lower in rural areas (35.6% in 2008) compared with urban areas (55.6% in 2008) in addition to policy emphasis on enhancing access to ECE in rural areas [49]. While administrative data are not available to estimate the changes in urban–rural and eastern–central–western gaps in kindergarten enrolment rates, nationwide survey data indicate that urban–rural and regional disparities in kindergarten attendance decreased significantly after 2010 [46].

Although there is a need for more official data on process quality, there has been an increased focus since 2010 by researchers on assessing both process quality and the structural factors that affect it. Research has consistently found that urban kindergartens outperform rural kindergartens on standardised instruments of process quality, including those measuring the provision of learning materials, activities, teacher–child interactions, and curriculum [47,50]. Teachers in urban ECE classrooms were reported to be more capable of providing children with stimulating learning environments, promoting children’s skills development through high-quality activities and interactions and responding to individual children’s needs [47]. These disparities in ECE quality may result in substantial gaps in children’s developmental outcomes.

Research conducted in China has also revealed that class size strongly affects teachers’ beliefs about developmentally appropriate activities and related practices [51,52]. Teachers leading a large group of children tend to spend more time on traditional whole-group instruction, meaning there is less time for individual attention to support young children’s learning [51]. Teachers with more education and those responsible for a smaller group of children are more likely to hold child-centred beliefs and implement high-quality instruction than other teachers [53].

The literature has also documented the association between kindergarten providers and process quality. In general, public kindergartens scored higher on structure and process quality indicators [50]. Furthermore, as public kindergartens run by educational departments generally receive more supervision from the local governments, they tend to have higher process quality than those run by non-educational departments and other public sectors.

### 5.2. Equity in Children’s Early Learning Outcomes and Development

As mentioned earlier, unlike at other levels in the Chinese education system, standardised tests are not used to assess children when they transition from kindergarten to school. Instead, kindergarten teachers may use the developmental outcomes outlined in the guidelines as an observational tool to evaluate the level of children’s developmental competencies and decide whether a child needs extra support. There is no nationwide data on young children’s learning outcomes based on adult reports or direct assessments of children. However, research has consistently reported that children in rural areas and those from lower-SES families perform at a lower level than their advantaged peers on standardised tests of early development [54,55].

While the evidence on the long-term impact of kindergarten participation on later achievement is inconclusive, it is promising. A study with a representative sample from five provinces in China further found that an additional nine months of kindergarten was positively associated with higher scores of cognitive development and social-emotional competence [50]. Interestingly, it also showed that an extra four hours of ECE was negatively associated with children’s social-emotional scores [56]. This was assumed be due to receiving less individual attention, for an extended time, because of group-based care. Gong and colleagues examined non-cognitive skills among rural adolescents [57]. They found that students who had attended kindergartens had more friends and were more likely to hold leadership positions in schools than their peers who did not participate in kindergarten. However, kindergarten participation was not related to these rural children’s literacy and mathematics abilities [57].

In China, there has been a lack of research on whether attending kindergarten benefits children from socioeconomically disadvantaged backgrounds more than children from other backgrounds. However, limited extant research has highlighted the compensatory effects of high-quality ECE for children living under socioeconomically disadvantaged circumstances. Li and colleagues examined the moderating influences of urbanicity and maternal education on the relationship between kindergarten quality and children’s language, early math, and social cognition in Zhejiang, a developed province in eastern China [55]. They found that the role of ECE quality was similar for children with high or low-educated mothers and whether living in urban or rural areas [55]. This provides evidence that ECE participation has universally positive role for children regardless of their family background. 

Given the paucity of nationally representative data on the relation between kindergarten experience and children’s learning outcomes, it is challenging to determine whether kindergarten expansion has successfully narrowed SES-based gaps in child outcomes. Considering the overall increases in indicators of structural quality over the years, one would anticipate improvements in the process quality of kindergartens, which may, in turn, be related to improved children’s learning outcomes. Meanwhile, it should be kept in mind that these linkages may be observed due to many other unmeasured structural elements of kindergarten quality and process quality also affect child development. More importantly, kindergartens serving children living in socioeconomically disadvantaged circumstances may need more support and more competent teachers to bring these children onto an equal footing with their more advantaged peers. However, in reality, these kindergartens may have limited positive impact due to their low quality. Relevant data must be collected at the national level to better evaluate and monitor ECE quality and its impact, identify disparities amongst children, and evaluate the impact of policy interventions [58].

## 6. Challenges and Opportunities

China has made tremendous efforts to improve access to high-quality ECE for children from varying backgrounds, showcasing the importance of the state in funding allocation, policymaking and regulation, and governing non-state actors. Despite the outstanding achievement in enhancing access to ECE, challenges remain in ensuring universal and equitable access to high-quality ECE. Longstanding challenges continue to hinder children’s development, including persistent regional and urban–rural disparities, challenges specific to migrant children, and an insufficient number of professionally trained teachers. These challenges are currently the focus of policy efforts, and the use of technology will further support the monitoring and regulation of ECE system quality. Free and compulsory ECE is yet to be legally mandated. We believe that when passed into law, the ECE law (draft) will signal a new era of ECE development in China. This will build on the remarkable developments that have occurred since 2010. An inclusive, accessible, affordable, and equitable ECE system is being developed to meet the United Nations Sustainable Development Goals (SDGs) targets that relate to early childhood development. The nation strives for all children to be afforded an opportunity to achieve their learning potential with the joint efforts of teachers, parents, and community members. An empowered, professional, and skilful ECE teaching workforce is the key to the success of an ECE system. More policy attention should be accorded to the well-being of ECE teachers, including adequate remuneration, increased job security, better working conditions, and diverse pathways to professional development and career progression.

## Figures and Tables

**Figure 1 children-10-01674-f001:**
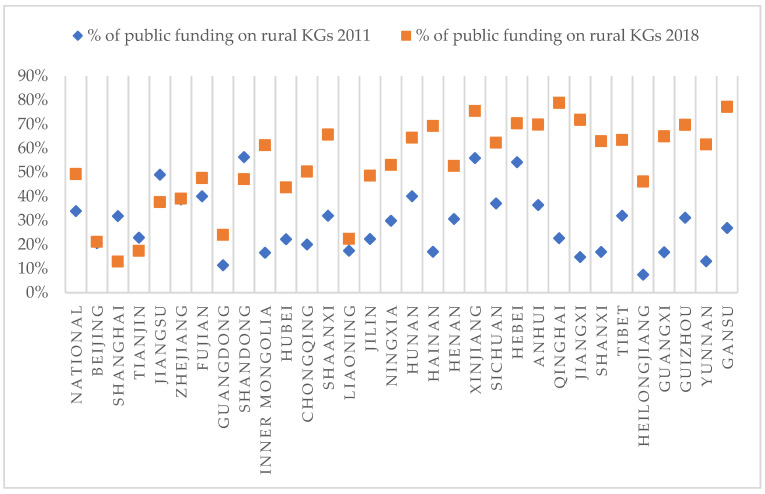
Trends in ECE public funding on rural kindergartens. Sources: Department of Finance, Ministry of Education (MoE), and Department of Social Science, Technology and Cultural Industry Statistics, NBS, 2013, 2019.

**Figure 2 children-10-01674-f002:**
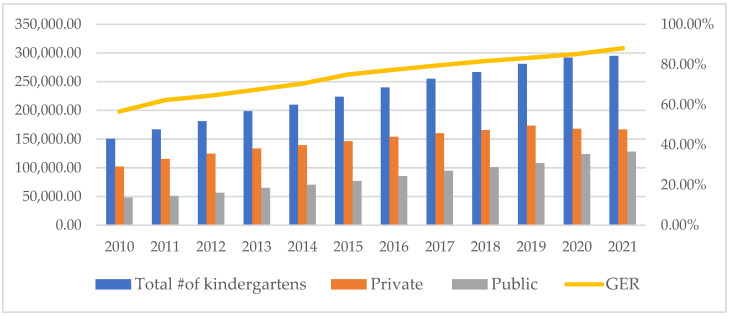
Number of kindergartens and GER (2010–2021). Source: MoE, 2011–2022.

## Data Availability

No new data were created or analysed in this study. Data sharing does not apply to this article.

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
