# Peer review of "Promoting Equity in Access to Quality Early Childhood Education in China"

_children, 2023, doi:10.3390/children10101674_

Round 1
Reviewer 1 Report
Thank you for an informative text about the development qualitatively and quantitatively in Peoples Republic of China. It was enlightening, easy to read, and a contribution for me to understand ECE policy in China.
But please check the following two lines. I think there are some typos there:
Line 52 2000 this the urbanization rate was 36%.
Lines 613-614 Free and compulsory ECE is yet to be? legally mandated.
I wish you luck with your publication.
Reviewer 2 Report
Good afternoon.
I have added some comments in the attached PDF.
Best regards.

Reviewer 3 Report
Title:
Promoting Equity in Access to Quality Early Childhood Education in China
The reviewer’s comments
The subject matter of this theme is good and well worth pursuing. However, the reviewer would like to see some revisions made to your manuscript.
1. The abstract will be revised to include details pertaining to various aspects such as data collection, participants, methodologies employed, and the instruments utilised.
2. The introduction should include problem context, literature review and the hypothesis based on the gap analysis of the previously published research. Significance of the study should be elaborated in further depth. Justify
3. A detailed critique of recent studies should be in the Literature Review.
4. Methodology is well elaborated and detailed. However please elaborate the theory or framework used to carry out particular enquiries.
5. Conclusion is elaborated as per requirement.
6. Review again after major revision.

Title:
Promoting Equity in Access to Quality Early Childhood Education in China
The reviewer’s comments
The subject matter of this theme is good and well worth pursuing. However, the reviewer would like to see some revisions made to your manuscript.
1. The abstract will be revised to include details pertaining to various aspects such as data collection, participants, methodologies employed, and the instruments utilised.
2. The introduction should include problem context, literature review and the hypothesis based on the gap analysis of the previously published research. Significance of the study should be elaborated in further depth. Justify
3. A detailed critique of recent studies should be in the Literature Review.
4. Methodology is well elaborated and detailed. However please elaborate the theory or framework used to carry out particular enquiries.
5. Conclusion is elaborated as per requirement.
6. Review again after major revision.
Reviewer 4 Report
Thank you to the authors for their research in this area and their work on this paper. I found it really interesting to review. I would recommend some further fine-tuning which primarily focuses on the increased integration of literature to substantiate some key points. I would also recommend that in some sections, the discussion/citations broaden out to international sources focusing on quality in ECE or the significance of ECE in general. This will help with contextualising and will create greater richness and depth. I also think in this space it is important to spend some time reflecting on the wellbeing of teachers in the ECE workforce - not only in terms of how they are coping/surviving/thriving in their work in psychosocial terms, but also in terms of how we support the workforce around continued professional learning.
Re: this line - "With the recognition of the importance of ECE for human and societal development, the Chinese government has increased investment in ECE" - I think it would help to spend some more time (even if just a paragraph) covering the importance of ECE in a focused way and with some depth & detail.
Re: lines 286-287 - "Second, the environment plays a vital role in children’s learning and development within 286 traditional Chinese philosophy" - this is also true in literature that covers the concept of the environment as the third teacher. I recommend further exploration of this literature to flesh out this point whether in the opening section or in the subsequent sections.
Re: section 4.1, especially "Class size, teacher-child ratios, and staff qualifications" - I think more integration of literature is needed here. You could look across international sources covering these kinds of issues and then relate back to the Chinese context. I think some international contextualisation may help to enrich this section, and perhaps a handful of others.
Re: lines 388-389 - "Globally, there is increasing consensus that a strong monitoring and quality assurance system is pivotal to the sustainable development of ECE systems [29]" - I would recommend more critical & comprehensive discussion around this point with reference to a wider range of literature. I think there are some interesting tension points to explore, even if only via a paragraph or so, as increased monitoring can impact on educator wellbeing if they are not provided with appropriate resources, supports, etc. Sometimes the push for 'quality' can create undue stress so these factors need to be balanced, especially given the reciprocal relationship between educator wellbeing and child wellbeing.
Re: line 611 - "and insufficient professionally trained teachers" - I think the wording here needs some minor adjustment, e.g. "and an insufficient number of professionally trained teachers". I would recommend some further writing on what can be done to upskill the workforce and care for their professional and personal wellbeing.
Re: lines 616 - 618: "We believe that the ECE law (draft) when passed into law will signal a 614 new era of ECE development in China which builds on the remarkable developments that 615 have occurred since 2010." I think this is a valid point to conclude but I would recommend that the conclusion be further developed with another sentence or two that help to wrap everything up and speak to your vision for high quality ECEC in China. This is a good opportunity to revisit and foreground key points as to what is needed.
Overall, a very interesting paper that explores the issues quite comprehensively. Well done to the authors and I wish them all the best with their future research.
Some minor copy editing required.
Reviewer 5 Report
This article provides an interesting overview of ECE in China. However, my main concern is that it focuses on policy developments, demographics and structural features only. What I am much more interested in is what actually is being "taught" in Chinese ECE institutions. Numerous times "high quality" is being used, but what this actually is is not made clear at al. It seems to me that this mainly pertains to structural features.
Therefore, the authors should write much more on the actual contents of ECE in China. Is there a national, state curriculum? Who decides what's the contents, and what is this contents? Does it differ from curricula in Western countries? How?
I understand that it is forbidden to teach literacy (l. 366-367) as well as having an "academic orientation". Then what is allowed to being "taught"? What is it ECE teachers actually do?
How many hours per week are children normally in ECE?
Are there special programs or interventions for disadvantaged children?
Are there differences between China and Hong Kong?
How much is Y170 billion in US Dollars (l. 163)?
Round 2
Reviewer 3 Report
Title:
Promoting Equity in Access to Quality Early Childhood Education in China
The reviewer’s comments
Thanks to the author for the correction. Revisions or explanations are all made according to the suggestions of the reviewers. Accept in present form.

Title:
Promoting Equity in Access to Quality Early Childhood Education in China
The reviewer’s comments
Thanks to the author for the correction. Revisions or explanations are all made according to the suggestions of the reviewers. Accept in present form.
Author Response
Thank you for your comments which helped us improve the manuscript.
Reviewer 4 Report
Thank you to the authors for their further work on the paper and their response to the feedback. The revisions have improved the paper and I feel it is almost ready for publication. One thought that I had was the following could use some expansion - this is an interesting addition and I would like to see it stay in the final paper, but maybe with some further writing to really expand on the meaning & significance? A little more explanation and discussion around this would be lovely:
"One of the best examples to illustrate how 333 Chinese philosophy emphasizes the importance of environment comes from the legend 334 of Mencius. The mother of Mencius, a widow, moved her residence three times until she 335 found the right location to raise her child. This was next door to a school."
Overall, a really interesting paper which has been strengthened through careful revision. Thank you again to the authors for their work on this manuscript and in this area of research.
Some minor editing required prior to publication.
Author Response
Thank you for the reviewer's suggestion. We have added discussion toward the legend of Mencius on pages 7-8, lines 448-467. See below.
One of the best examples to illustrate how Chinese philosophy emphasizes the importance of environment comes from the legend of Mencius. The mother of Mencius, a widow, moved her residence three times until she found the right location to raise her child. This was next door to a school. A favorable environment is assumed to promote proper and moral behaviors, while a chaotic and disorganized environment does not lead to propriety and effective learning. Thus, the kindergarten quality rating guidelines emphasize a structured learning environment with stimulating learning materials. Notably with the gradual improvement of kindergarten structural quality, the Chinese government has introduced specific guidelines and regulations to promote process quality.
Reviewer 5 Report
Thanks for providing extra information.
Author Response

(The authors gave the same response as above.)
